# MRI Delta Radiomics to Track Early Changes in Tumor Following Radiation: Application in Glioblastoma Mouse Model

**DOI:** 10.3390/biomedicines13040815

**Published:** 2025-03-28

**Authors:** Mohammed S. Alshuhri, Haitham F. Al-Mubarak, Abdulrahman Qaisi, Ahmad A. Alhulail, Abdullah G. M. AlMansour, Yahia Madkhali, Sahal Alotaibi, Manal Aljuhani, Othman I. Alomair, A. Almudayni, F. Alablani

**Affiliations:** 1Radiology and Medical Imaging Department, College of Applied Medical Sciences, Prince Sattam Bin Abdulaziz University, Alkharj 11942, Saudi Arabia; a.alhulail@psau.edu.sa (A.A.A.); ag.alqahtani@psau.edu.sa (A.G.M.A.); m.aljuhani@psau.edu.sa (M.A.); a.almudayni@psau.edu.sa (A.A.); f.alablani@psau.edu.sa (F.A.); 2Department of Radiology, Northwestern University, Chicago, IL 60611, USA; haitham.almubarak@northwestern.edu; 3Department of Radiology, Security Forces Hospital, Riyadh 11564, Saudi Arabia; ahq1401@gmail.com; 4Department of Diagnostic Radiography Technology, College of Nursing and Health Sciences, Jazan University, Jazan 45142, Saudi Arabia; ymedkhali@jazanu.edu.sa; 5Radiological Sciences Department, College of Applied Medical Sciences, Taif University, Taif 21944, Saudi Arabia; dahhasi@gmail.com; 6Faculty of Health & Life Sciences, University of Liverpool, Liverpool L69 7ZA, UK; 7Radiological Sciences Department, College of Applied Medical Sciences, King Saud University, P.O. Box 145111, Riyadh 4545, Saudi Arabia; oalomir@ksu.edu.sa

**Keywords:** delta radiomics, glioblastoma, radiation therapy, MRI, machine learning, tumor morphology and texture analysis

## Abstract

**Background/Objectives**: Glioblastoma (GBM) is an aggressive and lethal primary brain tumor with a poor prognosis, with a 5-year survival rate of approximately 5%. Despite advances in oncologic treatments, including surgery, radiotherapy, and chemotherapy, survival outcomes have remained stagnant, largely due to the failure of conventional therapies to address the tumor’s inherent heterogeneity. Radiomics, a rapidly emerging field, provides an opportunity to extract features from MRI scans, offering new insights into tumor biology and treatment response. This study evaluates the potential of delta radiomics, the study of changes in radiomic features over time in response to treatment or disease progression, exploring the potential of delta radiomics to track temporal radiation changes in tumor morphology and microstructure. **Methods**: A cohort of 50 female CD1 nude mice was injected intracranially with G7 glioblastoma cells and divided into irradiated (IR) and non-irradiated (non-IR) groups. MRI scans were performed at baseline (week 11) and post-radiation (weeks 12 and 14), and radiomic features, including shape, histogram, and texture parameters, were extracted and analyzed to capture radiation-induced changes. The most robust features were those identified through intra-observer reproducibility assessment, ensuring reliability in feature selection. A machine learning model was developed to classify irradiated tumors based on delta radiomic features, and statistical analyses were conducted to evaluate feature feasibility, stability, and predictive performance. **Results**: Our findings demonstrate that delta radiomics effectively captured significant temporal variations in tumor characteristics. Delta radiomics features exhibited distinct patterns across different time points in the IR group, enabling machine learning models to achieve a high accuracy. **Conclusions**: Delta radiomics offers a robust, non-invasive method for monitoring the treatment of glioblastoma (GBM) following radiation therapy. Future research should prioritize the application of MRI delta radiomics to effectively capture short-term changes resulting from intratumoral radiation effects. This advancement has the potential to significantly enhance treatment monitoring and facilitate the development of personalized therapeutic strategies.

## 1. Introduction

Glioblastoma (GBM) is the most aggressive and lethal primary brain tumor in adults, characterized by rapid growth, high infiltration into surrounding brain tissue, and resistance to standard therapies. Despite advances in surgical resection, radiotherapy, and chemotherapy, the prognosis for GBM remains poor, with a median survival of approximately 15 months following diagnosis [1]. The highly heterogeneous nature of GBM presents a significant challenge in treatment planning and response assessment, as tumors exhibit substantial variability between patients (interpersonal heterogeneity) and within a single tumor (intrapersonal heterogeneity) due to diverse genetic mutations, microenvironmental factors, and cellular differentiation. This heterogeneity influences tumor progression, immune evasion, and therapeutic resistance, making it difficult to predict treatment outcomes and personalize therapy effectively [2].

MRI is the preferred imaging modality for diagnosing GBM, planning surgeries, and monitoring therapeutic responses. MRI provides a superior soft tissue contrast and enables non-invasive assessments of tumor microstructure and heterogeneity. The availability of multiple imaging contrasts (e.g., T1-weighted, T2-weighted, diffusion-weighted imaging (DWI), magnetization transfer imaging, and chemical exchange saturation transfer (CEST)) enhances the ability to visualize the tumor environment at a voxel-by-voxel level, allowing for a more comprehensive evaluation of tumor evolution. However, despite these advantages, the assessment of treatment response in brain tumors remains limited by current MRI-based evaluation criteria, which primarily rely on tumor size measurements rather than capturing the full biological complexity of tumor progression.

Radiomics has emerged as a promising approach to enhance GBM imaging analysis by extracting high-dimensional features from MRI scans that describe tumor heterogeneity, texture, shape, and intensity variations [3,4,5,6]. These quantitative imaging biomarkers provide a more objective, reproducible, and comprehensive assessment of tumor behavior compared to conventional methods [7,8,9,10,11]. However, traditional radiomics captures imaging features at a single time point, failing to reflect temporal changes in tumor characteristics that occur during therapy.

To overcome this limitation, delta radiomics incorporates time-dependent feature analysis, allowing for the evaluation of tumor evolution over multiple imaging time points. Unlike traditional radiomics, delta radiomics considers variations in imaging features between multiple MRI scans acquired at different time points, enabling a longitudinal assessment of tumor evolution during treatment. By tracking therapy-induced modifications, delta radiomics offers a more precise, patient-specific evaluation of tumor response, which can aid in treatment adaptation and personalization.

Delta radiomics has demonstrated promising results in treatment planning, understanding tumor growth patterns, and monitoring therapeutic responses; however, there is a scarcity of studies assessing its effectiveness relative to single-time-point radiomics. Zhang et al. [12] reported that delta features derived from two follow-up MR images of brain metastases after radiosurgery, when analyzed using an ensemble classifier model, achieved an accuracy of 73.2% and an AUC of 0.73 in distinguishing between radiation necrosis and tumor progression. This study investigated changes in radiomics features during radiation therapy for non-small-cell lung cancer and their impact on prognosis, observing significant changes, with delta radiomics improving survival and metastasis predictions. Despite its potential, the integration of delta radiomics into clinical workflows remains a major challenge. The methodologies for calculating and interpreting voxel-by-voxel changes in tumor parameters are poorly standardized, computationally intensive, and not widely understood. Furthermore, while high-resolution MRI and advanced signal processing methods have facilitated histogram-based tumor analysis, the use of delta radiomics for assessing tumor radiosensitivity and predicting treatment response remains underexplored.

This study aims to utilize delta radiomics features at different time points to evaluate the effects of early radiation therapy in a preclinical mouse model of glioblastoma. By leveraging MRI-based radiomic analysis, we aim to characterize temporal changes in tumor morphology, texture, and heterogeneity following radiation treatment. The study will focus on longitudinal feature extraction, allowing for a more detailed assessment of tumor evolution across multiple imaging time points. Additionally, we will apply machine learning techniques to classify irradiated (IR) tumors at different stages based on delta radiomic features. To our knowledge, there are no published studies that have explored MR delta radiomics in brain tumors, nor ones that have examined lesion response to radiation with different endpoints.

The goal is to develop predictive models that can identify patterns of tumor response to radiation therapy, providing insights into treatment efficacy and resistance mechanisms. By using AI analysis, we seek to establish imaging biomarkers that correlate with therapeutic effectiveness, which could eventually be translated into clinical applications for glioblastoma and other brain tumors.

## 2. Materials and Methods

### 2.1. Animal Model

Fifty female CD1 nude mice were intracranially injected with G7 glioblastoma cells and divided into the following two groups: irradiated (IR, n = 42) and non-irradiated controls (non-IR, n = 8). Tumors were allowed to grow for 11 weeks before MRI scanning (Figure 1). The mice were monitored twice daily for signs of distress, including >20% weight loss, lethargy, neurological deficits, or feeding abnormalities, and were euthanized if they met humane endpoint criteria using CO_2_ asphyxiation followed by cervical dislocation. All procedures were performed under isoflurane anesthesia, with buprenorphine (0.05 mg/kg SC) for analgesia. The mice were housed in controlled conditions with a 12 h light/dark cycle, ad libitum food and water, and nesting material for enrichment. All 50 mice were euthanized based on humane endpoints. The study was approved by the Institutional Animal Ethics Committee (REC-HSD-205-2023) at Prince Sattam Bin Abdulaziz University and followed the ARRIVE guidelines and NC3Rs principles.

### 2.2. MRI Imaging

MRI experiments were conducted using a 7 Tesla Bruker Biospec Avance system (Bruker Biospin, Ettlingen, Germany). Homogeneous radiofrequency excitation was achieved with a 72 mm birdcage volume resonator, while signal detection was performed using an actively decoupled 4-channel phased array receive-only head surface coil (Rapid Biomedical, Wurzburg, Germany). The mice were initially anesthetized with 5% isoflurane and a 30:70 O_2_/N_2_O ratio, positioned prone on an MRI cradle. A hot water circulation jacket regulated the animal’s temperature at 37 ± 1 °C, monitored via a rectal probe. The head was secured laterally with conical ear rods and longitudinally with a nose cone for anesthetic gas delivery. The animals breathed spontaneously through a facemask that delivered a constant flow of isoflurane mixed with a 40:60 ratio of O_2_/N_2_O (1 L/min^−1^). The isoflurane concentration was adjusted between 1.5% and 3% to maintain stable respiration rates within normal physiological ranges (40–70 bpm). Respiration was continuously monitored throughout the experiment using a pressure sensor connected to an air-filled balloon placed under the animal’s abdomen (Topspin software 3.1, Bruker, Ettlingen, Germany, https://www.bruker.com/en/products-and-solutions/mr/nmr-software/topspin.html, accessed on 24 March 2025).

MRI was conducted at the following three time points: week 11 (pre-radiation baseline), week 12, and week 13. Post-radiation MRI scans were also conducted in week 13 to assess tumor progression and the effects of radiation therapy. T2-weighted rapid acquisition with relaxation enhancement (RARE) sequences were used to obtain high-resolution images with the following parameters: repetition time (TR) of 2500 ms, echo time (TE) of 33 ms, field of view (FOV) of 25 mm × 25 mm, slice number (n = 8), resolution of (176 × 176) pixels, and thickness of 0.5 mm. These settings ensured the optimal visualization of tumor boundaries, enabling accurate tumor segmentation for radiomic analysis.

### 2.3. Irradiation Procedure

Targeted radiation was delivered using the Small Animal Radiation Research Platform (SARRP), which allows for precise irradiation with beam sizes as small as 0.5 mm, minimizing radiation exposure to surrounding normal tissues. The SARRP system (Xstrahl, Version 5.0) uses cone-beam computed tomography (CBCT) for guidance, ensuring the precise localization of the radiation beam within the brain, mimicking clinical treatment protocols. The IR group received a fractionated radiation dose of 6 Gy, delivered in three 2 Gy fractions over three consecutive days. This fractionation scheme was chosen to minimize acute toxicity while delivering an effective therapeutic dose. The entire brain was irradiated to account for the diffuse nature of GBM tumors, which exhibit highly infiltrative growth patterns.

### 2.4. Radiomics Features Extraction and Selection

T2-weighted MRI images were used for radiomics analysis. Tumor regions of interest (ROIs) were manually segmented using 3D Slicer (version 5.2.1) by two observers with 10 years of experience in neuroimaging analysis. Care was taken to exclude surrounding normal brain tissue to ensure accurate tumor segmentation by the two observers after applying a normalizing image filter. The segmentation pipeline is illustrated in Figure 2. Following segmentation, a total of 107 radiomic features were extracted from each MRI scan, capturing tumor morphology (14), texture (75), and intensity variations (18). Feature extraction was performed using 3D slicer, ensuring compliance with the Image Biomarker Standardization Initiative (IBSI). Additionally, inter-class correlation coefficient (ICC) calculations were performed to extract robust radiomics features assessed by the two observers, which were categorized based on their reliability, where a good reliability was defined as ICC ≥ 0.8, a moderate reliability as ICC = 0.5–0.8, and a poor reliability as ICC < 0.5. Only features with a good reliability (n = 58, Appendix A) were retained for further analysis. The radiomics processing pipeline, including segmentation and feature extraction, is depicted in Figure 2.

### 2.5. Delta Radiomics Feature Analysis

To capture temporal changes in tumor characteristics, delta radiomics was applied by analyzing variations in radiomic features across multiple time points (week 11, with post-radiation follow-ups in week 12 and week 13). Changes in radiomic features over time were computed using *p*-values of <0.05. By quantifying changes in feature distributions over time, delta radiomics provides a longitudinal assessment of treatment response, allowing us to identify key biomarkers that distinguish responders from non-responders to radiation therapy.

### 2.6. Machine Learning-Based Predictive Modeling

To classify tumors as irradiated (IR) at the time points of week 11, 12, and 13 to assess radiation response, machine learning models were implemented using Python (3.12.7). Delta radiomics features were used to distinguish between IR in weeks 11, 12, and 13 to ensure robustness and generalizability after applying the z score. Then, 5-fold cross-validation was conducted. The models were assessed by the area under the receiver operating characteristic curve.

### 2.7. Statistical Analysis

Statistical analyses were performed using MATLAB R2024b to identify robust radiomic features with interobserver concordance correlations (ICCs > 0.8). A two-sided Wilcoxon signed-rank test was employed to extract and compare variations in delta radiomic features across different time points. Graphs were created using GraphPad Prism 10, while Python code was utilized for training, testing, and evaluating the machine learning models.

## 3. Results

### 3.1. Radiomics Features Analysis

The analysis revealed significant differences in the shape, histogram, and texture features between the irradiated (IR) and non-irradiated (non-IR) groups. The shape features in the IR group, particularly the major axis length and compactness, showed a marked reduction in week 14 compared to the non-IR group, indicating tumor shrinkage after radiation therapy. Other shape metrics, such as spherical disproportion, which quantifies how much a three-dimensional shape deviates from a perfect sphere, commonly used in medical imaging analysis in radiomics to describe the irregularity of tumors or other anatomical structures and volume density, were also significantly different in the IR group, capturing the morphological impacts of radiation (*p* < 0.05 for all shape features) (see Table 1, shape features).

Histogram features showed substantial variations as well, with the IR group displaying a decrease in global intensity peak, mean intensity, and intensity range, suggesting changes in tumor density and cellularity. In particular, global and local intensity peaks, along with the interquartile intensity range, exhibited statistically significant declines (*p* < 0.05) between the IR and non-IR groups at weeks 1 and 14. These changes reflect radiation-induced necrosis and a reduction in viable tumor cells, which alter the intensity distribution within the tumor region (see Table 1, histogram features).

Texture features also indicated distinct patterns between the IR and non-IR groups. Notably, features such as Gray-Level Non-Uniformity, Dependence Entropy, and Gray-Level Variance were significantly higher in the IR group, capturing the increased heterogeneity within the tumor matrix post-radiation. This aligns with expected biological responses, as radiation causes cell death and disrupts tumor structure, leading to a more heterogeneous texture profile (*p* < 0.05) (see Table 2, texture features).

### 3.2. Tumor Growth Analysis Pre- and Post-Irradiation

The analysis of tumor growth over time utilized T2-weighted MRI images, with tumors being manually delineated. At baseline (week 11), there were no significant differences (*p*-value < 0.05) in tumor volume measured by 3D analysis between the non-irradiated (non-IR, mean = 13.37 ± 5.95) and irradiated (IR, mean = 13.82 ± 9.12) groups, indicating initial equivalence. However, by weeks 12 and 13 post-irradiation, non-significant differences in tumor growth were observed at these time points, as follows: in week 12, the non-IR group had a mean of 30.71 ± 16.17, while the IR group had a mean of 27.74 ± 22.64; in week 13, the non-IR group had a mean of 24.64 ± 6.72, compared to the IR group with a mean of 39.86 ± 29.07 (see Figure 3).

### 3.3. Quantitative Analysis

The radiomics features assessed by two observers in the pre-radiation groups (n = 47) were analyzed using the intraclass correlation coefficient (ICC) to identify robust features (ICC > 0.8, n = 55) for subsequent analysis, as shown in Table 1. The robust radiomics features extracted from the T2-weighted images exhibited high ICCs, including shape (78.56%), histogram (72.22%), GLCM (62.5%), GLDM (42.86%), GLRLM (50%), GLSZM (12.5%), and NGTDM (60%). Moderate ICCs were observed for shape (21.43%), histogram (27.78%), GLCM (25%), GLDM (57.14%), GLRLM (43.75%), GLSZM (68.75%), and NGTDM (40%). Poor ICCs were recorded for shape (0%), histogram (0%), GLCM (12.5%), GLDM (0%), GLRLM (6.25%), GLSZM (18.75%), and NGTDM (%0), as shown in Figure 4.

The findings indicated that most shape, histogram, GLDM, and NGTDM features exhibited high and moderate ICC values in the pre-radiation groups for both non-IR and IR. These features also displayed a good variability, making them suitable for subsequent analysis steps.

### 3.4. Delta Radiomics Features in IR and Non-IR Groups

To assess the statistical significance of differences between the IR and non-IR groups at different time points, a *p*-value analysis was calculated by MATLABR2024b (Wilcoxon method) for each radiomics feature. The non-IR group in weeks 11, 12, and 13 was not statically significant at all time points. The IR group in week 11 and week 12 (n = 53) was non-statistically significant (shape features 10%, histogram 18.18%, and texture 2.7%) and statistically significant (shape features 90%, histogram 81.81%, and texture 97.29%) (Figure 5a), and the IR group in week 12 (n = 47) and week 13 was non-statistically significant (shape features 10%, histogram 27.27%, and texture 18.91%) and statistically significant (shape features 90%, histogram 72.72%, and texture 81%) (Figure 5b).

### 3.5. Machine Learning Classification and PCA

For classification purposes, machine learning models were employed with a 70–30 split for training and testing data. Principal Component Analysis (PCA) was applied to the extracted MRI radiomics features after Z-score normalization. The first two principal components were selected, accounting for 95% of the total variance, to reduce dimensionality and retain the most informative features, leading to a robust set of predictive features used in the machine learning model. Support vector machine (SVM) and logistic regression models were applied and distinguished IR from weeks 11 and 12 and IR from weeks 12 and 13, which allowed for effectively identifying the structural and intensity-based changes induced by radiation, as shown in Table 1.

The AUC results demonstrated the model’s overall reliability in differentiating between irradiated (IR) samples at week 11 versus week 12 and week 12 versus week 13 (Figure 6). Collectively, these metrics confirm the effectiveness of integrating radiomics features with machine learning to accurately evaluate treatment effects in this glioblastoma (GBM) model.

## 4. Discussion

This study’s findings underscore the capability of MRI-based delta radiomics to detect notable temporal alterations in glioblastoma (GBM) morphology and microstructure post-radiation therapy. Prior radiomics studies have typically analyzed single time points, lacking the temporal dimension needed to capture dynamic tumor changes. Delta radiomics addresses this gap by examining how radiomic features evolve across multiple imaging time points, providing a longitudinal view of tumor progression. Unlike static radiomics, which may miss transient treatment-induced effects, delta radiomics tracks radiation-related changes in tumor heterogeneity, shape, and texture over time. This dynamic approach could potentially offer a more precise and patient-specific assessment of tumor response, informing adaptive treatment strategies and identifying robust imaging biomarkers for early therapeutic effects. By employing delta radiomics in our GBM mouse model, we introduce a promising methodology to detect subtle microstructural changes post-radiation that traditional imaging might overlook.

Our observations align with and extend prior radiomics research. For instance, Núñez et al. (2020) demonstrated that radiomic texture and intensity features from MRIs could distinguish chemotherapy-treated vs. control GBM in mice [13]. While their study was cross-sectional and specific to temozolomide (with a different GBM model), both their work and ours highlight that imaging features capture subtle therapy-induced changes, such as an altered heterogeneity. Notably, the specific delta features we identified differ (reflecting radiation’s unique effects versus chemotherapy), yet both studies underscore heterogeneity measures as key indicators of treatment effects. Our approach using delta radiomics provides additional insights beyond static radiomics by revealing how these features evolve, a capability that static, single-time-point analyses lack. Moreover, our findings echo the notion expressed by Hooper and Ginat (2023) that radiomics augments conventional MRI by yielding quantitative markers linked to genotype, response, and prognosis [14]. However, delta radiomics offers further granularity, capturing early temporal shifts that may forecast longer-term outcomes. This dynamic approach has been lauded as a tool for early therapy adaptation in radiation oncology. By comparing our results with these studies, we demonstrate that delta radiomics not only corroborates known radiomic signatures of treatment response, but also provides a richer, time-resolved perspective that static radiomics cannot, thereby enhancing the ability to personalize and adjust treatments earlier in the care pathway. A notable observation of this study was the diminished voxel intensity uniformity and increased tumor heterogeneity in the IR group following radiation therapy. These findings align with prior research highlighting the significance of radiomics in delineating intra-tumoral heterogeneity, which has been associated with tumor growth and treatment resistance [15]. The utilization of delta radiomics enabled us to monitor these alterations over time, offering a more dynamic and thorough evaluation of tumor response in contrast to traditional static imaging methods. The integration of machine learning models improved our capacity to distinguish between irradiated and non-irradiated tumors at different time intervals, thereby validating the efficacy of AI-driven radiomics in treatment assessment [16].

Ensuring the robustness of radiomic feature selection was a critical aspect of our methodology. Initially, we identified features with a high reproducibility in intra-observer analyses, minimizing variability from image acquisition and segmentation differences. This approach ensured that only reliable and stable features were selected for further analysis. Subsequently, we focused on dynamic delta features that captured temporal variations in tumor characteristics. By utilizing these comprehensive delta radiomic features, we enhanced the sensitivity of our analysis in detecting radiation-induced changes in tumor morphology and texture [17]. This methodological framework adheres to established principles in radiomics research, ensuring that our findings are statistically significant and reproducible.

Despite these promising outcomes, several challenges remain in the clinical application of delta radiomics. A constraint is the absence of uniformity in radiomic feature extraction and machine learning pipelines, which can impact reproducibility and generalizability. Our research employed rigorous feature selection techniques and strong quality control protocols to ensure the dependability of the derived radiomic characteristics. Nonetheless, as previously addressed in systematic reviews of delta radiomics, multicenter validation and standardized methods are crucial for enhancing clinical translation [17]. Moreover, although our machine learning models showed a commendable accuracy in distinguishing between the IR and non-IR groups, additional validation in larger, independent datasets is essential to verify their predictive efficacy. Another factor is the biological interpretation of radiomic characteristics. Although radiomics offers quantitative metrics of tumor form and texture, the fundamental biological mechanisms responsible for these alterations have yet to be completely clarified. Integrating radiomic analysis with supplementary genetic and histopathological data may provide an enhanced understanding of the biological associations of radiomic features. Prior research has underscored the necessity for multimodal strategies that integrate radiomics with genomes, transcriptomics, and pathology to enhance the predictive efficacy of imaging biomarkers [13,14]. In our study, the entire brain was irradiated (as opposed to focal beam targeting) due to the highly infiltrative nature of G7 glioblastoma tumors, which lack clear borders. This approach ensured that radiation was delivered to all regions of tumor cell infiltration and avoided under-dosing microscopic disease. We acknowledge that whole-brain irradiation is different from clinical practice (where focal radiotherapy is standard), and it may induce changes in normal brain tissue. However, whole-brain irradiation is often employed in preclinical GBM studies for consistency when dealing with diffuse tumor models [18]. The precision of the SARRP system was utilized to confine the doses to the cranium and minimize off-target exposure. We also chose to exclude chemotherapy in this study to isolate radiation-induced effects on the MRI features; adding temozolomide (TMZ) would make it difficult to discern whether radiomic changes were due to radiation or the drug. We agree that future experiments should explore focal irradiation and chemo–radiotherapy combinations to more closely mimic clinical treatment and assess how such factors might alter radiomic signatures. Another key limitation of this study is the use of immunodeficient (nude) mice, which lack functional T-lymphocytes. While this was necessary to permit human GBM cell implantation, the absence of an adaptive immune response means that we could not capture immune-related effects on tumor growth or radiomic features. This limitation likely precludes observations of radiation-induced inflammatory or immunological changes in the MRI data. Future studies should consider immunocompetent or humanized GBM models to assess how the immune system’s involvement may influence radiomic patterns after therapy. Additionally, our use of a 9.4T small-animal MRI scanner differs from the 3T scanners commonly used in clinical settings, which may affect the translatability of our findings. Investigating the performance of these radiomic markers at clinical field strengths is essential for potential clinical application.

## 5. Conclusions

In summary, this study demonstrates that MRI-based delta radiomics provides a dynamic, sensitive measure of GBM tumor response following radiation. By capturing temporal shifts in tumor morphology and texture, delta radiomics identified early microstructural alterations that preceded visible size changes. This approach addresses a key gap in traditional imaging by offering time-resolved insights into treatment efficacy. Our delta radiomic features, validated with machine learning, successfully differentiated irradiated from non-irradiated tumors across time points, underscoring the promise of AI-driven radiomic models in oncology. While challenges in standardization, reproducibility, and clinical translation remain, our findings contribute to the growing evidence that radiomics can serve as a non-invasive biomarker for early treatment monitoring. With continued research to validate and refine this methodology, delta radiomics could be integrated into clinical practice to guide personalized GBM treatment, adapting therapies in real time to improve patient outcomes. The robust and dynamic nature of delta radiomics marks a significant advancement in our ability to non-invasively track, predict, and ultimately improve cancer therapy responses.

## Figures and Tables

**Figure 1 biomedicines-13-00815-f001:**
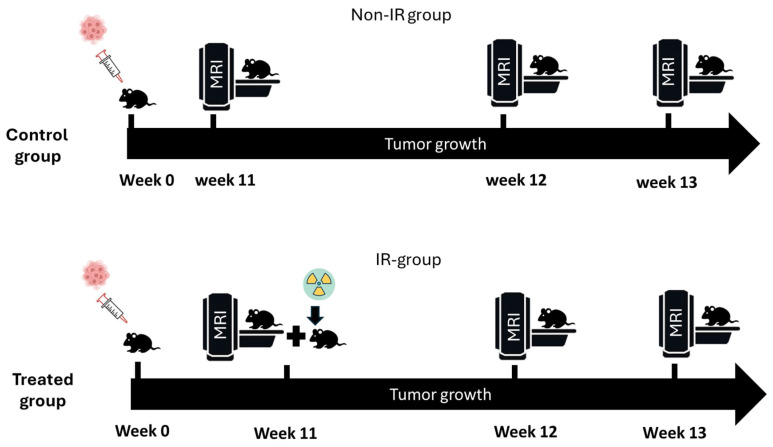
Experimental protocol. At week zero, 50 mice were orthotopically implanted with G7 glioblastoma cells. Mice were divided into two groups (non-IR and IR) and were imaged in different time points (weeks 11, 12, and 13).

**Figure 2 biomedicines-13-00815-f002:**
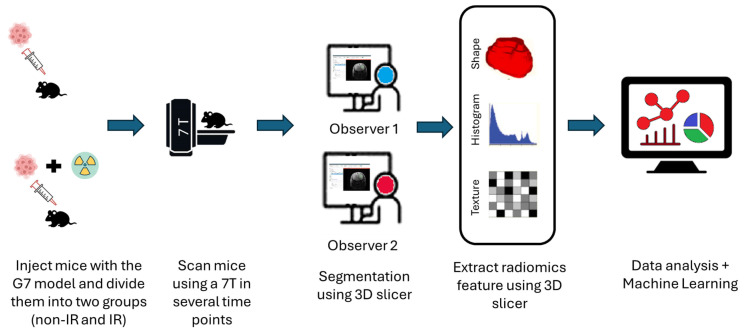
Schematic diagram of the radiomics analysis pipeline steps after injecting the G7 model. The procedures include image acquisition, applying normalization image filter, tumor segmentation, radiomic features (shape, histogram, and texture) extraction, and then delta radiomics features analysis using predictive model construction and validation.

**Figure 3 biomedicines-13-00815-f003:**
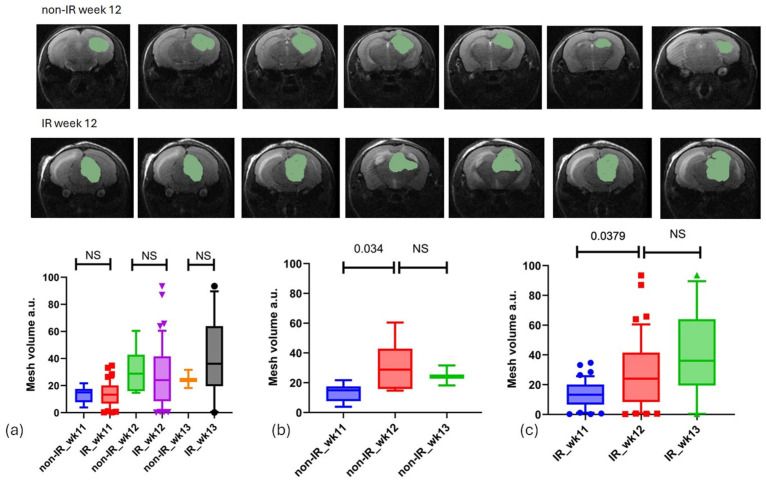
Tumor regions were manually outlined from T2-weighted slices where they were visible. (**a**) Illustrates tumor growth for both the IR and non-IR groups across three time points (weeks 11, 12, and 13), revealing no significant difference between the two groups (unpaired *t*-test). (**b**,**c**) Present a comparison of tumor growth between the non-IR and IR groups at various time points.

**Figure 4 biomedicines-13-00815-f004:**
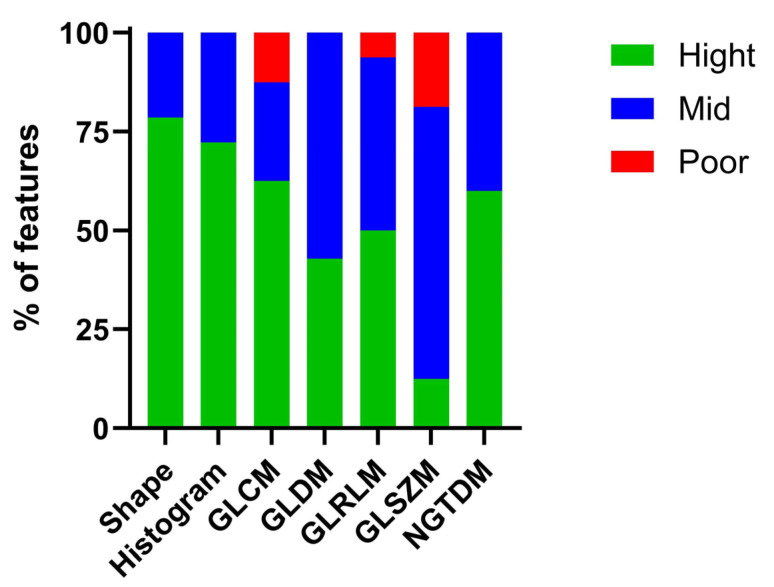
Shows the percentage of radiomics features extracted from inter observers between pre-radiation groups, high ICC values (ICCs > 0.8) were selected as robust radiomics features.

**Figure 5 biomedicines-13-00815-f005:**
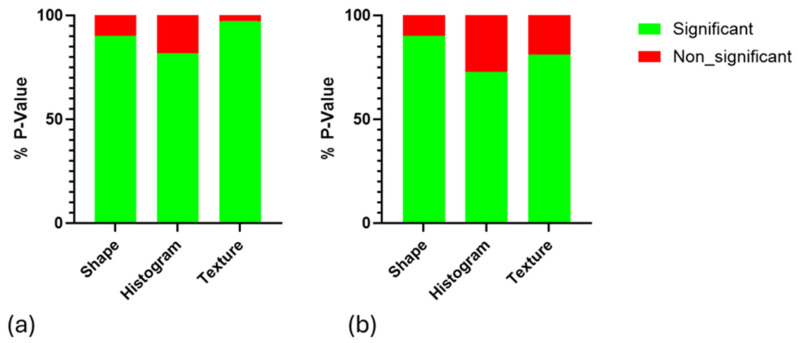
Illustrating the *p*-value analysis (*p* < 0.05) of delta radiomic features across different IR groups at various time points: (**a**) week 11 compared to week 12 and (**b**) week 12 compared to week 13.

**Figure 6 biomedicines-13-00815-f006:**
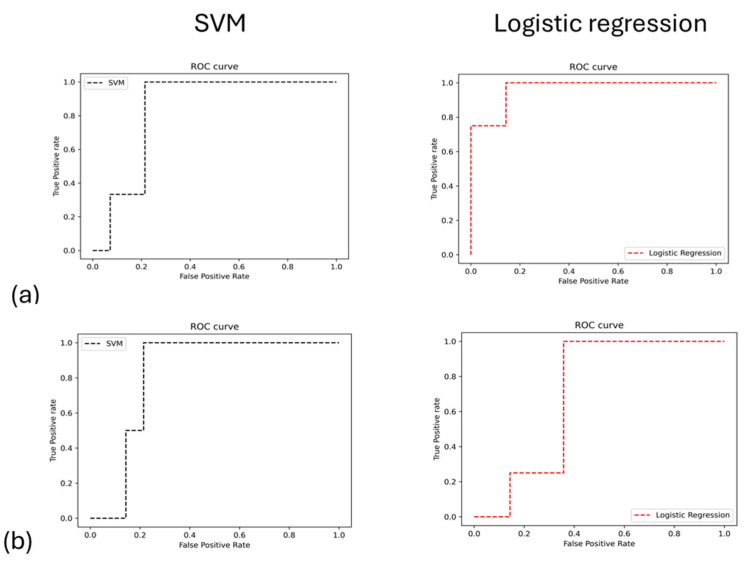
The analysis of ROC performance for machine learning models is conducted between two sets of IR groups: (**a**) during weeks 11 and 12 and (**b**) during weeks 12 and 13.

**Table 1 biomedicines-13-00815-t001:** Delta radiomics features that demonstrate significant differences (*p*-values < 0.05) between the intervention groups in weeks 11, 12, and 13.

IR (Week 11 VS. Week 12)	IR (Week 12 VS. Week 13)	Category
Elongation	Elongation	shape
LeastAxisLength	LeastAxisLength	shape
Maximum2DDiameterColumn	Maximum2DDiameterColumn	shape
Maximum2DDiameterRow	Maximum2DDiameterRow	shape
Maximum3DDiameter	Maximum3DDiameter	shape
MeshVolume	MeshVolume	shape
MinorAxisLength	MinorAxisLength	shape
SurfaceArea	SurfaceArea	shape
SurfaceVolumeRatio	SurfaceVolumeRatio	shape
VoxelVolume	VoxelVolume	shape
×10Percentile	×10Percentile	histogram
×90Percentile	×90Percentile	histogram
Energy	Energy	histogram
Entropy	Entropy	histogram
InterquartileRange	InterquartileRange	histogram
Mean	Mean	histogram
Median	Median	histogram
Minimum	---------------	histogram
RootMeanSquared	RootMeanSquared	histogram
TotalEnergy	TotalEnergy	histogram
ClusterProminence	ClusterProminence	histogram
ClusterShade	ClusterShade	GLCM
ClusterTendency	ClusterTendency	GLCM
Contrast	Contrast	GLCM
Correlation	Correlation	GLCM
DifferenceAverage	DifferenceAverage	GLCM
DifferenceEntropy	DifferenceEntropy	GLCM
DifferenceVariance	DifferenceVariance	GLCM
	Idn	GLCM
Imc1	Imc1	GLCM
Imc2	Imc2	GLCM
InverseVariance	InverseVariance	GLCM
JointEnergy	JointEnergy	GLCM
JointEntropy	JointEntropy	GLCM
SumEntropy	SumEntropy	GLCM
SumSquares	SumSquares	GLCM
DependenceNonUniformity	DependenceNonUniformity	GLDM
GrayLevelNonUniformity	GrayLevelNonUniformity	GLDM
GrayLevelVariance	GrayLevelVariance	GLDM
LargeDependenceEmphasis	LargeDependenceEmphasis	GLDM
SmallDependenceEmphasis	SmallDependenceEmphasis	GLDM
	LargeDependenceEmphasis	GLDM
SmallDependenceHighGrayLevelEmphasis	---------------	GLDM
GrayLevelNonUniformity_1	GrayLevelNonUniformity_1	GLDM
GrayLevelVariance_1	---------------	GLDM
LongRunEmphasis	LongRunEmphasis	GLDM
RunEntropy	RunEntropy	GLDM
RunPercentage	---------------	GLDM
RunVariance	RunVariance	GLDM
ShortRunEmphasis	---------------	GLDM
ShortRunHighGrayLevelEmphasis	ShortRunHighGrayLevelEmphasis	GLDM
LargeAreaEmphasis	LargeAreaEmphasis	GLSZM
LargeAreaLowGrayLevelEmphasis	LargeAreaLowGrayLevelEmphasis	GLSZM
Busyness	Busyness	NGTDM
Complexity	Complexity	NGTDM
Contrast_1	---------------	NGTDM

**Table 2 biomedicines-13-00815-t002:** Presents the performance metrics of various machine learning models, including Random Forest, Decision Tree, and Logistic Regression, evaluated across training, testing, and AUC (area under the curve) measures.

Model Name	Training Accuracy %		Testing Accuracy %		AUC	
	IR wk 11 vs. wk 12	IR wk 12 vs. wk 13	IR wk 11 vs. wk 12	IR wk 12 vs. wk 13		
SVM	86	77	74	78	0.83	0.82
Logistic regression	77	74	77	72	0.96	0.7

## Data Availability

There are no restrictions to the data presented in this article.

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
