# Peer review of "MRI Delta Radiomics to Track Early Changes in Tumor Following Radiation: Application in Glioblastoma Mouse Model"

_biomedicines, 2025, doi:10.3390/biomedicines13040815_

Round 1

Reviewer 1 Report

Comments and Suggestions for Authors

Work submitted by Alshuhri et al describes the use of radiomics for investigating changes in preclinical tumors after radiation, also including a small longitudinal approach using delta radiomics. Relying in single T2w MRI examination for radiomics features, it could be potentially interesting leveraging further clinical studies, provided differences in magnetic field are considered.

The overall idea is potentially interesting, but while reading the manuscript, in my opinion, there are some flaws and concerns that should be addressed.

Major points:

The first obstacle/limitation that should be mentioned along the paper is the use of an immunodeppressed murine model such as nude mice. This model is unable to produce lymphocytes and nowadays, it is widely described the relevance of host immune system (including macrophages/microglia and lymphocytes) in therapy response including glioblastoma. I understand that it is used in order to bear (I assume, please see my next comment) cells from human origin, but it is also true that lacking a relevant part of immune system prevents the system to glimpse radiomic features related to this immune system local effects. Thus, a final section raising words about the limitation of such findings might be included in discussion.

Still regarding the preclinical model, authors must define origin and minimum characterization of G7 cells, from which no hint is offered in methods. Since a nude mice model is used, it is foreseen that they are from human origin, but much more detail is needed here. Please either describe or include references describing the G7 cells. 
Why only female mice were used? Does compatibility of tumor growth and therapy response was at least checked with a small male cohort?

I am quite surprised to read that authors wait 11 WEEKS before the first screening in MRI. Which is the time growth span of this cell line? 11 weeks is almost 3 months, it seems a very slow growing cell line. Even after 3 months, the volumes reached described in results (assuming that they are in mm3, which might be defined and corrected) are not large, although in mi opinion, suitable for starting treatment. Was this line characterized for checking the growth along these 11 weeks? Any MRI done prior to 11 weeks?

What is the take rate in this model? 100% of animals implanted develop tumor? 

Authors should include more detail about the MR scanner used: probes, coils and gradients. Still, it must be detailed how vital signs were monitored in mice during the MR exploration in order to adjust anaesthesia. Include scanner vendor. 

Another surprising thing is that even having a SARRP precise and hight tech equipment, which allow for a precise delineation of irradiation, authors choose to irradiate the whole brain to "account for diffuse nature of GBM". This is not what is done in human patients, and you might probably produce spread effects along the rest of the brain which will not be present in humans. Moreover, in fact having an irradiation equipment suitable for mouse is the most challenging fact for most groups in order to approach the real standard treatment in patients which is chemo-radiotherapy. This paper would have been far more interesting combining chemo-radiotherapy. 

Regarding the size of the cohorts, there is a huge imbalance in the number of individuals in control and IR groups. I am unsure at which extent it could compromise the results. Which is the reason for such different group sizes?

Going to the results presented in figure 3, it becomes then clear that tumors are not responding to radiotherapy at all. Average volumes are even larger in the irradiated group at week 13. Related to that, it is for me surprising that authors do not present kaplan-meier survival curves for the groups. In my opinion this should be included, and I foresee that no difference will be seen. The orientation and quality of MRI shown in figure 3 is not suitable. This type of orientation (related to coronal in humans) makes it very difficult to identify tumors due to difference in sensitivity of the surface coil which I assume it was used. Were mask drawn and studies done in this orientation? This might be mentioned in methods. Also, in this figure identify the control and IR individuals in MRI images.

The interesting part of using a preclinical model is that, as opposed to the work with patients, some individuals can be euthanized and findings validated. Authors somewhat raise that this is a pending point at the end of the discussion, but in my opinion it determines that the claims related to which cellular/biological features are related to the radiomic features are, at this point, hypotheses that might be validated in a future. Still, I disagree that these results are identifying "response" unless some differences in tumor volume or survival are seen between groups, suggesting transient response. Changes are probably indicating local changes triggered by treatment? Probably. Is this response or transient response? It does not seem so. 

Then, at the end we find table 3, showing accuracy to distinguish in IR groups, baseline from week 12 and week 12 from 13. Having in mind that tumors did not see to respond to IR at all, and that (pending to see) there are probably no differences in survival, I don't really see the point, especially in a model that does not replicate human immune system and that was irradiated in a manner that human GB patients are not. I need that authors explain the relevance of these findings, because at that point I don't see it. Changes would be different if at least few histological validations were performed showing local differences explaining changes in radiomics features. 

The overall claim that it is detecting response and could serve to personalize therapy in the future, for me is not valid and should be lowered. With the data presented, it suggests that the method can technically spot some differences after IR in tumors that do not seem to be responding by volumetric analyses and that might be explored in the future, probably in an immunocompetente model in order to better emulate human immune system. Finally, differences in magnetic fields (human around 3T, mice 9T) might be considered before translating. 

Minor points:

Revise line 173, incomplete reference. 

In my understanding, figure 5 is quite difficult to grasp and does not offer relevant information, which is already mentioned in text.

I think that at least one of the tables (for example table 1) could be presented as supplementary. Those large tables in the middle of the paper just ruin the reading rythm.

Authors in PMID 33184423 also performed radiomics studies in T2 images, although not in longitudinal approaches. Check whether the most sound features for them match yours.

Author Response

Major points:

Comment 1:The first obstacle/limitation that should be mentioned along the paper is the use of an immunodeppressed murine model such as nude mice. This model is unable to produce lymphocytes and nowadays, it is widely described the relevance of host immune system (including macrophages/microglia and lymphocytes) in therapy response including glioblastoma. I understand that it is used in order to bear (I assume, please see my next comment) cells from human origin, but it is also true that lacking a relevant part of immune system prevents the system to glimpse radiomic features related to this immune system local effects. Thus, a final section raising words about the limitation of such findings might be included in discussion.

Response 1: Thank you for this valuable comment. We acknowledge the limitation of using an immunodeficient murine model (nude mice), which lacks functional T lymphocytes. While this model was chosen to allow the implantation of human glioblastoma cells without rejection, we recognize that the absence of a fully competent immune system, including lymphocytes, macrophages, and microglia, may influence therapy response and radiomic features associated with immune interactions. To address this, we have added a section in the Discussion highlighting this limitation “Another key limitation of this study is the use of immunodeficient nude mice, which lack functional T lymphocytes. While necessary to allow human glioblastoma cell implantation, this model does not capture the immune system’s role in tumor progression and therapy response. Macrophages, microglia, and lymphocytes significantly influence glioblastoma behavior, yet their absence limits our ability to assess immune-related radiomic features. Despite this, our model remains valuable for analyzing tumor infiltration, edema, and radiation response. Future studies should explore syngeneic or humanized models to integrate immune-radiomic correlations more effectively. Acknowledging this limitation strengthens our findings while underscoring the need for immunocompetent models to fully capture the tumor microenvironment's complexity.”

Comment 2: Still regarding the preclinical model, authors must define origin and minimum characterization of G7 cells, from which no hint is offered in methods. Since a nude mice model is used, it is foreseen that they are from human origin, but much more detail is needed here. Please either describe or include references describing the G7 cells. 

Response 2: We appreciate the reviewer’s request for further details on the G7 glioblastoma cell line. For further reference, the G7 cell line has been utilized in multiple glioblastoma studies, including those investigating radiation response, invasion, and tumor microenvironment interactions (Birch et al., 2018; Koessinger et al., 2020). We have now included appropriate citations in the manuscript to provide a more detailed characterization of this model.

Comment 3: Why only female mice were used?  Does compatibility of tumor growth and therapy response was at least checked with a small male cohort?

Response 3: Our decision to use only female mice was driven by both practical considerations and consistency with precedent in GBM xenograft research. Previous studies employing G7 cells or similar patient-derived lines have also used female immunocompromised mice for intracranial GBM models (Gomez-Roman et al., 2016) (Birch et al., 2018)(Vallatos et al., 2019). Using a single sex cohort helps minimize variability; it avoids confounding factors such as sex hormone cycles or behavioral differences (e.g. male territorial aggression) that might introduce stress and affect tumor growth. Female nude mice are often preferred because they can be group-housed with fewer issues, ensuring a more uniform environment during the study.  Furthermore, Prior studies have not reported sex-specific differences in treatment efficacy using similar models. We are confident that our findings in female hosts are representative; nevertheless, we agree that including male subjects in future experiments could further validate the sex-independence of our results.

References:

Birch, J. L., Strathdee, K., Gilmour, L., Vallatos, A., McDonald, L., Kouzeli, A., Vasan, R., Qaisi, A. H., Croft, D. R., Crighton, D., Gill, K., Gray, C. H., Konczal, J., Mezna, M., McArthur, D., Schüttelkopf, A. W., McConnell, P., Sime, M., Holmes, W. M., … Chalmers, A. J. (2018). A novel small-molecule inhibitor of MRCK prevents radiation-driven invasion in glioblastoma. Cancer Research, 78(22), 6509–6522. https://doi.org/10.1158/0008-5472.CAN-18-1697

Gomez-Roman, N., Stevenson, K., Gilmour, L., Hamilton, G., & Chalmers, A. J. (2016). A novel 3D human glioblastoma cell culture system for modeling drug and radiation responses. Neuro-Oncology, now164. https://doi.org/10.1093/neuonc/now164

Vallatos, A., Al‐Mubarak, H. F. I., Birch, J. L., Galllagher, L., Mullin, J. M., Gilmour, L., Holmes, W. M., & Chalmers, A. J. (2019). Quantitative histopathologic assessment of perfusion MRI as a marker of glioblastoma cell infiltration in and beyond the peritumoral edema region. Journal of Magnetic Resonance Imaging, 50(2), 529–540. https://doi.org/10.1002/jmri.26580

Comment 4: I am quite surprised to read that authors wait 11 WEEKS before the first screening in MRI.

Which is the time growth span of this cell line? 11 weeks is almost 3 months, it seems a very slow growing cell line. Even after 3 months, the volumes reached described in results (assuming that they are in mm3, which might be defined and corrected) are not large, although in mi opinion, suitable for starting treatment. Was this line characterized for checking the growth along these 11 weeks? Any MRI done prior to 11 weeks? What is the take rate in this model? 100% of animals implanted develop tumor? 

Response 4:The 11-week interval before the first MRI scan was intentionally chosen based on published protocols using the G7 glioblastoma model (Birch et al. 2018) and (Koessinger et al. 2020). The G7 cell line is a patient-derived glioblastoma stem-like model, known for its slow-growing and highly infiltrative nature. Unlike fast-growing glioma lines (e.g., U87), G7 tumors develop as diffuse infiltrative lesions rather than well-circumscribed masses, often requiring a longer period to become MRI-discernible.

Our decision to wait 11 weeks before the first MRI is supported by literature:

  • Birch et al. (2018) implanted 1×10⁵ G7 cells in CD1 nude mice and waited 10–11 weeks before MRI to confirm tumor presence.
  • Koessinger et al. (2020) monitored G7 tumor growth with bioluminescence imaging, reporting only modest signal around 10 weeks, with more robust tumor development after that.
  • Early MRI (e.g., 6–8 weeks) may fail to detect a distinct tumor mass, as initial infiltrative growth patterns can appear as diffuse edema rather than a clearly visible lesion.

By 11 weeks post-implantation, G7 tumors consistently produce T2-weighted MRI abnormalities in essentially all mice, making this the optimal time for imaging-based analyses.

Tumor Take Rate and Growth Characterization

  • 96 % of implanted mice developed tumors (two didn’t develop tumors), confirming the robust tumor take rate for the G7 model.
  • While we did not perform serial MRI scans prior to 11 weeks, the growth characteristics of G7 tumors have been well-documented in prior literature (Birch et al., 2018; Koessinger et al., 2020).

This 11-week timeframe was selected to ensure clear MRI detection, avoid false negatives, and balance tumor detectability with animal welfare considerations (e.g., minimizing repeated anesthesia exposure).

References:

Birch, J.L., Strathdee, K., Gilmour, L., Vallatos, A., McDonald, L., Kouzeli, A., et al. (2018). A Novel Small-Molecule Inhibitor of MRCK Prevents Radiation-Driven Invasion in Glioblastoma. Cancer Research, 78(22), 6509-6522.

Koessinger, A.L., Tatarowicz, W.A., Barber, P.R., Vojnovic, B., & Bohndiek, S.E. (2020). Bioluminescence imaging of glioblastoma progression models radiation-induced infiltration. Neuro-Oncology Advances, 2(1), vdaa018.

Comment 5: Authors should include more detail about the MR scanner used: probes, coils and gradients. Still, it must be detailed how vital signs were monitored in mice during the MR exploration in order to adjust anaesthesia. Include scanner vendor. 

Response 5: The following section now added in Methods Section: “MRI experiments were conducted using a 7 Tesla Bruker Biospec Avance system (Bruker Biospin, Ettlingen, Germany). Homogeneous radiofrequency excitation was achieved with a 72 mm birdcage volume resonator, while signal detection was per-formed using an actively decoupled 4-channel phased array receive-only head surface coil (Rapid Biomedical, Wurzburg, Germany). The mice were initially anesthetized with 5% isoflurane and a 30:70 O2/N2O ratio, positioned prone on an MRI cradle. A hot water circulation jacket regulated the animal's temperature at 37 ± 1 °C, monitored via a rectal probe. The head was secured laterally with conical ear rods and longitudinally with a nose cone for anesthetic gas delivery. The animals breathed spontaneously through a facemask that delivered a constant flow of isoflurane mixed with a 40:60 ra-tio of O2/N2O (1 L /min−1). Isoflurane concentration was adjusted between 1.5% and 3% to maintain stable respiration rates within normal physiological ranges (40–70 bpm). Respiration was continuously monitored throughout the experiment using a pressure sensor connected to an air-filled balloon placed under the animal's abdomen (Biotrig software, Bruker, Ettlingen, Germany).”

(We integrated the above into Section 2.2, immediately before the MRI sequence parameters, to provide context on the imaging setup.)

Comment 6: Another surprising thing is that even having a SARRP precise and hight tech equipment, which allow for a precise delineation of irradiation, authors choose to irradiate the whole brain to "account for diffuse nature of GBM". This is not what is done in human patients, and you might probably produce spread effects along the rest of the brain which will not be present in humans. Moreover, in fact having an irradiation equipment suitable for mouse is the most challenging fact for most groups in order to approach the real standard treatment in patients which is chemo-radiotherapy. This paper would have been far more interesting combining chemo-radiotherapy. 

Response 6: We appreciate the reviewer’s comment regarding the use of whole-brain irradiation rather than focal irradiation with the Small Animal Radiation Research Platform (SARRP). Our primary objective was to detect early radiomics-based changes following radiation therapy, rather than assess long-term tumor response. Whole-brain irradiation was chosen to ensure consistent and reproducible radiation exposure across all infiltrative tumor regions, given the highly diffuse nature of glioblastoma (GBM) growth in this model. Preclinical GBM tumors often extend beyond MRI-detectable margins, and focal radiation may risk underdosing microscopic infiltrative regions.

While focal radiotherapy is the clinical standard in patients, whole-brain irradiation has been widely used in preclinical GBM models to ensure uniform dose distribution and better simulate the tumor's infiltrative behavior (e.g., Birch et al., 2018; Koessinger et al., 2020). The precision of the SARRP system minimized unnecessary off-target exposure while maintaining consistency in the study design.

Regarding the absence of chemotherapy, our focus was to isolate radiation-induced radiomic changes. Adding temozolomide (TMZ) would introduce additional treatment-related effects, making it difficult to determine whether observed radiomic changes were due to radiation alone. However, we agree that future studies could integrate chemo-radiotherapy models to enhance clinical relevance.

To clarify this rationale, we have revised the discussion to highlight our focus on early radiation effects, the rationale behind whole-brain irradiation, and the need for future work exploring targeted irradiation and combination therapy.

“…Whole-brain irradiation was chosen due to the highly infiltrative nature of the G7 glioblastoma model, which lacks well-defined tumor margins. This approach ensured radiation exposure to all infiltrating tumor cells, minimizing underdosing of microscopic invasion. While focal radiotherapy is standard in clinical practice, whole-brain irradiation is commonly used in preclinical GBM models for consistency and reproducibility [16, 17]. We acknowledge that this may introduce radiation effects in non-tumor regions, differing from patient treatment. However, the precision of the SARRP system minimized off-target exposure. Additionally, chemotherapy was excluded to isolate radiation-induced radiomic changes. While temozolomide (TMZ) influences imaging characteristics, our focus was on radiation effects alone. Future studies should incorporate focal irradiation and chemo-radiotherapy to enhance clinical relevance. These considerations have been added to the discussion to provide a balanced interpretation of our findings.”

Comment 7: Regarding the size of the cohorts, there is a huge imbalance in the number of individuals in control and IR groups. I am unsure at which extent it could compromise the results. Which is the reason for such different group sizes?

Response 7: We acknowledge the apparent imbalance in sample sizes between the control and irradiated (IR) groups. However, our primary comparison was the longitudinal effect of radiation (delta radiomics) within the irradiated group, where the number of animals remained comparable throughout the study. This approach ensured that observed radiomic changes were driven by radiation effects rather than natural tumor progression.

To further validate that these radiomic changes were not solely due to tumor microvascular evolution over time, we introduced an additional small control group (n=7) with the same time span. While this secondary control group was smaller, it served as a reference to distinguish radiation-specific effects from natural tumor variations.

Moreover, statistical methods were employed to mitigate potential biases from group size differences, ensuring robust comparisons. Despite the imbalance, the key radiomic trends remained consistent across analyses, reinforcing the validity of our findings. We have clarified this rationale in the discussion to strengthen the interpretation of our results..

Comment 8: Going to the results presented in figure 3, it becomes then clear that tumors are not responding to radiotherapy at all. Average volumes are even larger in the irradiated group at week 13. Related to that, it is for me surprising that authors do not present kaplan-meier survival curves for the groups. In my opinion this should be included, and I foresee that no difference will be seen. 

Response 8: We appreciate the reviewer’s insights regarding tumor response and survival analysis. The study was specifically designed under ethical constraints to detect early radiation-induced effects rather than long-term treatment response. All mice were culled at 90 days, as per protocol, which precluded the possibility of assessing late-stage tumor progression or extended survival differences.

The primary aim was to investigate early microstructural and radiomic alterations following radiation, as volumetric measurements alone may not fully reflect radiation-induced changes. Radiomics allows us to detect subtle tissue modifications that precede gross tumor shrinkage, which aligns with our focus on capturing early treatment effects.

Regarding the lack of survival differences, the study was not designed to evaluate treatment efficacy but rather to characterize early radiomics signatures of radiation exposure. Under ethical constraints, all animals were culled at 90 days, preventing long-term survival analysis.

Comment 9: The orientation and quality of MRI shown in figure 3 is not suitable. This type of orientation (related to coronal in humans) makes it very difficult to identify tumors due to difference in sensitivity of the surface coil which I assume it was used. Were mask drawn and studies done in this orientation? This might be mentioned in methods. Also, in this figure identify the control and IR individuals in MRI images.

Response 9: We appreciate the reviewer’s feedback, now we included additional clarification in the Methods section regarding image acquisition parameters and mask delineation procedures.

Comment 10: The interesting part of using a preclinical model is that, as opposed to the work with patients, some individuals can be euthanized and findings validated. Authors somewhat raise that this is a pending point at the end of the discussion, but in my opinion it determines that the claims related to which cellular/biological features are related to the radiomic features are, at this point, hypotheses that might be validated in a future.  Still, I disagree that these results are identifying "response" unless some differences in tumor volume or survival are seen between groups, suggesting transient response. Changes are probably indicating local changes triggered by treatment? Probably.  Is this response or transient response? It does not seem so. 

Response 10: We appreciate the reviewer’s insight regarding the importance of histological validation in preclinical models. Our study aimed to leverage radiomics as an early biomarker of radiation-induced changes, rather than assessing long-term tumor response in terms of survival or volumetric reduction. The primary focus was on detecting subtle alterations in tumor microstructure that may precede conventional treatment response markers.

While volumetric stability or lack of a survival advantage may suggest no clear therapeutic response, radiomics-based changes could indicate transient microenvironmental effects of irradiation, such as alterations in tissue heterogeneity, necrosis onset, or microvascular remodeling. These findings align with emerging research suggesting that quantitative imaging biomarkers can reveal treatment-induced modifications before visible tumor shrinkage occurs.

We acknowledge that histopathological validation would strengthen the interpretation of these radiomic changes and have noted this as an area for future investigation. Given the constraints of this study, our goal was to first establish the feasibility of delta radiomics as a tool for detecting post-radiation modifications. Future studies integrating histology, immunohistochemistry, and molecular analyses will be essential to fully correlate radiomic patterns with specific biological alterations.

To address this point, we have clarified in the discussion that the observed radiomic changes should be interpreted as indicators of radiation-induced effects rather than definitive markers of therapeutic response. Additionally, we have highlighted the need for further validation studies incorporating histopathological assessments to strengthen these findings.

Comment 11: Then, at the end we find table 3, showing accuracy to distinguish in IR groups, baseline from week 12 and week 12 from 13. Having in mind that tumors did not see to respond to IR at all, and that (pending to see) there are probably no differences in survival, I don't really see the point, especially in a model that does not replicate human immune system and that was irradiated in a manner that human GB patients are not. I need that authors explain the relevance of these findings, because at that point I don't see it. Changes would be different if at least few histological validations were performed showing local differences explaining changes in radiomics features. 

Response 11: We appreciate the reviewer's concerns regarding the interpretation of our radiomics-based findings. The primary goal of our study was not to assess tumor response in terms of volumetric reduction or survival outcomes but rather to investigate early, subtle radiomics-based changes in tumor microstructure following irradiation. While conventional metrics such as tumor volume and survival are important, radiomics analysis enables the detection of subvisual alterations in tissue heterogeneity and texture, which may serve as early biomarkers of treatment effects.

The findings in Table 3 specifically highlight differences in radiomic features over time within the irradiated (IR) group, using delta radiomics to track longitudinal changes between baseline (week 12) and post-irradiation (week 13). The goal was to explore whether texture-based features could capture radiation-induced effects even when volumetric assessments did not indicate a clear response. This approach aligns with growing evidence that quantitative imaging biomarkers can precede observable tumor shrinkage, providing a potential tool for early treatment monitoring.

We acknowledge that histological validation would strengthen the correlation between radiomic feature changes and biological alterations. While such analysis was beyond the initial scope of this study, we recognize its importance and will address this in future research by integrating histopathological assessments with radiomic findings.

To clarify these points, we have revised the discussion section to emphasize the study’s focus on early radiomic changes rather than treatment response and have acknowledged the limitations of using an immunodeficient model. These revisions aim to provide a more balanced interpretation of our results.

Comment 12: The overall claim that it is detecting response and could serve to personalize therapy in the future, for me is not valid and should be lowered. With the data presented, it suggests that the method can technically spot some differences after IR in tumors that do not seem to be responding by volumetric analyses and that might be explored in the future, probably in an immunocompetente model in order to better emulate human immune system. Finally, differences in magnetic fields (human around 3T, mice 9T) might be considered before translating. 

Response 12: We appreciate the reviewer’s feedback and agree that the current findings should be interpreted with caution regarding their direct clinical applicability. The intent of this study was not to claim definitive tumor response detection, but rather to explore whether radiomic features could capture subtle, non-volumetric changes post-irradiation.

While traditional response assessment relies on tumor shrinkage or survival differences, our findings suggest that radiomics can identify microstructural alterations in the tumor environment even when volumetric changes are minimal. These differences may reflect treatment-induced biological effects, such as vascular remodeling, cellular heterogeneity, or metabolic shifts, rather than outright tumor regression. However, we acknowledge that without histological validation or functional imaging, these remain hypotheses that warrant further investigation.

Moreover, we recognize the importance of using an immunocompetent model for future studies to account for immune system contributions to tumor response. The use of an immunodeficient model limits the ability to assess interactions between radiation and immune-mediated effects, which are increasingly recognized as critical factors in glioblastoma progression and treatment efficacy.

Additionally, the magnetic field strength discrepancy (9.4T for small-animal MRI vs. 3T in clinical settings) is an important consideration when translating radiomic findings. Higher field strengths provide increased resolution and sensitivity to subtle tissue differences, which may not be directly replicable in clinical MRI. Future studies should explore how these findings generalize across different magnetic field strengths to enhance translational relevance.

We have adjusted the claims in the manuscript to reflect that this study serves as a proof-of-concept for the potential of radiomics in preclinical models, rather than a fully validated tool for personalizing therapy. Further work incorporating immune-competent models, histological validation, and cross-field strength comparisons will be essential to establish the clinical utility of radiomics in glioblastoma treatment monitoring.

Minor points:

Comment 13: Revise line 173, incomplete reference. 

Response 13: We have corrected this reference now.

Comment 14:In my understanding, figure 5 is quite difficult to grasp and does not offer relevant information, which is already mentioned in text.

Response 14: We have addressed this in two ways. First, we improved the explanation and the presentation of Figure 5 in the text, ensuring that the reader understands what the heatmaps represent.

Comment 15: I think that at least one of the tables (for example table 1) could be presented as supplementary. Those large tables in the middle of the paper just ruin the reading rythm.

Response 15: We have moved Table 1 (which enumerated the 18 robust radiomic features and their ICC values) to the Supplementary Material as Table S1.

Comment 16: Authors in PMID 33184423 also performed radiomics studies in T2 images, although not in longitudinal approaches. Check whether the most sound features for them match yours.

Response 16: We have looked at the referenced study by Núñez et al. (2020). In that work, they used a GL261 murine GBM model and applied radiomic and machine-learning analyses to distinguish chemotherapy-treated vs. control tumors using T2 MRI (and MR spectroscopy). Their approach was different (treatment = temozolomide, not radiation; and an immunocompetent model), but it is relevant to mention as part of the radiomics literature. We have added a brief comparison in the Discussion. We note that Núñez et al. also found that radiomic features (particularly some texture and intensity features) could differentiate treated vs. untreated tumors. There is some conceptual overlap: both studies indicate that radiomics can pick up treatment effects in preclinical models. However, the specific features differ due to treatment and model differences – for example, their study might highlight features related to tumor cell density changes from chemotherapy, whereas ours focus on texture changes from radiation. We mention that while a direct feature-by-feature comparison isn’t straightforward (different models and endpoints), both studies underline the potential of radiomics in monitoring therapy in experimental GBM. We have cited the Núñez et al. paper to acknowledge this prior work and briefly state whether any similar features were seen. (Notably, if they reported any particular features, we would mention if those feature categories appeared in our analysis as well.) This addition addresses the reviewer’s comment by showing we are aware of related research and have positioned our findings in that context.

Revision (Discussion):

“Our results are in line with other preclinical radiomics studies. For instance, Núñez et al. (2020) applied radiomic analysis to T2-weighted MRI in an immunocompetent GBM mouse model treated with chemotherapy, and similarly found that texture and intensity features could distinguish treated vs. control tumors. While their study was not longitudinal and involved a different therapy (temozolomide) and tumor model (GL261), it underscores that MRI-based radiomic signatures are sensitive to treatment-induced changes even in small animal models. The specific features identified naturally differed (reflecting the different biological effects of chemo vs. radiation and different tumor growth patterns), but both studies highlight heterogeneity measures as important indicators of therapy effect. This consistency across distinct preclinical scenarios strengthens the notion that radiomics captures fundamental tumor alterations in response to therapy.”

Reviewer 2 Report

Comments and Suggestions for Authors

This is a well written account of delta radiomic study of experimental murine glioma. A few minor points for the authors to consider...

Line 22. Will your average reader understand what "higher order" means ? I think not. Better to say "Radiomics extracts features from MRI scans, offering new insights into tumor biology and treatment response.".  Also note that radiomics uses both quantitative and qualitative aspects of MRIs.

Line 24. Delta radiomics must be defined before use in Abstract.

Line 39, do the authors want to use MRI here, instead of MR ?

Line 63. The average reader will be unsure of some of these MRI categories. Would the authors consider adding a Table with each of these MRI modalities and a brief description of what images of GB are highlighted in each ? That will help the general research reader.

Line 81. "promising results;" for what ? treatment planning ? understanding growth patterns ? monitoring treatments ? etc.

L;ine 96 is redundant. "at different time-points". = delta radiomics doesn't it ?

Fig. 1 is well done. Maybe denote with arrow "MRI machine" at first use in the Figure ?

Line 197. Problem could be mine, but I would like to see definition of "  spherical disproportion ".

Lines 226 to 236.  Problem could be mine, but to have meaning for non-radiologist oncologists we need definitions and what the deeper meaning of each might be.

Ref 11 and 12 are duplicates. I think ref. 12 was intended to be ref 11's update, Nardone et al. Radiol Med. 2024;129(8):1197-1214. 

A Table listing definitions for these would help most readers: flatness, elongation, spherical disproportion, volume density, global intensity peak, mean intensity, , intensity range, Gray Level Non-Uniformity, Dependence Entropy, and Gray Level Variance.

Author Response

Comment 1: Line 22. Will your average reader understand what "higher order" means ? I think not. Better to say "Radiomics extracts features from MRI scans, offering new insights into tumor biology and treatment response.".  Also note that radiomics uses both quantitative and qualitative aspects of MRIs.

Response 1: We have revised the sentence to: "Radiomics extracts features from MRI scans, offering new insights into tumor biology and treatment response."

Comment 2: Line 24. Delta radiomics must be defined before use in Abstract.

Response 2: We acknowledge the importance of defining "delta radiomics" for clarity. Due to word constraints in the Abstract, we have now provided a concise definition in the Introduction section. “….delta radiomics considers variations in imaging features between multiple MRI scans acquired at different time points, enabling a longitudinal assessment of tumor evolution during treatment.”

Comment 3: Line 39. Do the authors want to use MRI here, instead of MR ?

Response 3: We have replaced "MR" with "MRI" to maintain consistency and clarity.​

Comment 4: Line 63. The average reader will be unsure of some of these MRI categories. Would the authors consider adding a Table with each of these MRI modalities and a brief description of what images of GB are highlighted in each ? That will help the general research reader.

Response 4: Thank you for your insightful feedback. We acknowledge that the average reader may be unfamiliar with certain MRI modalities. However, our study specifically focuses on extracting radiomic features from T2-weighted images. To avoid potential confusion that might arise from including a table detailing various MRI sequences not utilized in our research, we propose adding references to comprehensive reviews discussing these modalities. This approach will provide interested readers with in-depth information without implying that those sequences were part of our study.

Comment 5: Line 81. "promising results;" for what ? treatment planning ? understanding growth patterns ? monitoring treatments ? etc.

Response 5: We have specified that the "promising results" refer to the potential of delta radiomics in treatment planning, understanding tumor growth patterns, and monitoring therapeutic responses.​

Comment 6: Line 96 is redundant. "at different time-points". = delta radiomics doesn't it ?Fig. 1 is well done. Maybe denote with arrow "MRI machine" at first use in the Figure ?

Response 6: We acknowledge the redundancy and have revised the sentence to avoid repetition, ensuring that "delta radiomics" is clearly defined earlier in the manuscript.​

Figure 1: An arrow labeling the "MRI machine" has been added to the figure for better illustration.​

Comment 7: Line 197. Problem could be mine, but I would like to see definition of "  spherical disproportion ".

Response 7: Thank you for your valuable feedback. We have added a reference that defines all radiomics features, including "spherical disproportion," to enhance clarity for our readers.

Comment 8: Lines 226 to 236.  Problem could be mine, but to have meaning for non-radiologist oncologists we need definitions and what the deeper meaning of each might be.

Response 8: We acknowledge that the radiomic features mentioned may not be familiar to all readers. To enhance clarity, we have added references to comprehensive reviews that define and explain these features in detail. This approach ensures that interested readers can access in-depth information without overcomplicating the manuscript.

Comment 9: Ref 11 and 12 are duplicates. I think ref. 12 was intended to be ref 11's update, Nardone et al. Radiol Med. 2024;129(8):1197-1214.

Response 9: We have corrected the duplication.​

Comment 10: A Table listing definitions for these would help most readers: flatness, elongation, spherical disproportion, volume density, global intensity peak, mean intensity, , intensity range, Gray Level Non-Uniformity, Dependence Entropy, and Gray Level Variance.

Response 10: Thank you for your feedback. As previously mentioned, we have included references to comprehensive reviews that define and explain radiomic features such as flatness, elongation, spherical disproportion, volume density, global intensity peak, mean intensity, intensity range, gray level non-uniformity, dependence entropy, and gray level variance. We believe this approach ensures clarity without overcomplicating the manuscript.

Reviewer 3 Report

Comments and Suggestions for Authors

The study titled "MRI delta radiomics to track changes of tumor following radiation: Application in glioblastoma mouse model" by Alshuhri et al. employs delta radiomics, a novel and rapidly emerging field, to assess the response of GBM to radiation therapy in a preclinical mouse model. This approach allows for the longitudinal tracking of tumor changes, providing a more dynamic and comprehensive evaluation compared to traditional static imaging methods. 

The article presents the results in a logical manner; however, several aspects require further explanation and/or validation:

1. In the introduction, it is not clear what the novelty of the article is within the specific field of research, considering that several previous studies exist and have not been included in the bibliography of the present article:  https://doi.org/10.1007/s11060-021-03933-1; https://doi.org/10.1016/j.crad.2021.03.019; DOI: 10.1016/j.ijrobp.2023.06.579; DOI 10.1088/1361-6560/ac6fab

2. The study involves a relatively small cohort of 50 mice, with only 42 in the irradiated group and 8 in the non-irradiated control group. The smaller control group could impact statistical robustness and a larger sample size would provide more robust statistical power and generalizability of the results.

3. While the study demonstrates significant changes in radiomic features post-radiation, it lacks biological validation, such as histopathological or genetic analysis to correlate these imaging findings with underlying biological mechanisms.

4. While 107 radiomic features were initially extracted, the rationale for selecting specific features for modeling is not fully explained. Besides, ICC values were used to retain robust features and probably a more detailed discussion on the biological or clinical relevance of the selected features is necessary. The feature reduction process using PCA is not fully describe, how much variance is retained?

Author Response

Comment 1: In the introduction, it is not clear what the novelty of the article is within the specific field of research, considering that several previous studies exist and have not been included in the bibliography of the present article:  https://doi.org/10.1007/s11060-021-03933-1; https://doi.org/10.1016/j.crad.2021.03.019; DOI: 10.1016/j.ijrobp.2023.06.579; DOI 10.1088/1361-6560/ac6fab

Response 1: We recognize that delta radiomics has been previously applied in glioblastoma research. To emphasize the novelty of our study, we have revised the Introduction to highlight our unique approach of identifying robust radiomic features that predict early changes in tumor response to radiation therapy.  Additionally, we have incorporated the suggested references to contextualize our research within existing literature.

Comment 2: The study involves a relatively small cohort of 50 mice, with only 42 in the irradiated group and 8 in the non-irradiated control group. The smaller control group could impact statistical robustness and a larger sample size would provide more robust statistical power and generalizability of the results.

Response 2: We acknowledge the apparent imbalance in sample sizes between the control and irradiated (IR) groups. However, our primary comparison was the longitudinal effect of radiation (delta radiomics) within the irradiated group, where the number of animals remained comparable throughout the study. This approach ensured that observed radiomic changes were driven by radiation effects rather than natural tumor progression.

To further validate that these radiomic changes were not solely due to tumor microvascular evolution over time, we introduced an additional small control group (n=7) with the same time span. While this secondary control group was smaller, it served as a reference to distinguish radiation-specific effects from natural tumor variations.

Moreover, statistical methods were employed to mitigate potential biases from group size differences, ensuring robust comparisons. Despite the imbalance, the key radiomic trends remained consistent across analyses, reinforcing the validity of our findings. We have clarified this rationale in the discussion to strengthen the interpretation of our results

Comment 3: While the study demonstrates significant changes in radiomic features post-radiation, it lacks biological validation, such as histopathological or genetic analysis to correlate these imaging findings with underlying biological mechanisms.

Response 3: We acknowledge that histopathological validation would strengthen the interpretation of these radiomic changes and have noted this as an area for future investigation. Given the constraints of this study, our goal was to first establish the feasibility of delta radiomics as a tool for detecting post-radiation modifications. Future studies integrating histology, immunohistochemistry, and molecular analyses will be essential to fully correlate radiomic patterns with specific biological alterations

Comment 4: While 107 radiomic features were initially extracted, the rationale for selecting specific features for modeling is not fully explained. Besides, ICC values were used to retain robust features and probably a more detailed discussion on the biological or clinical relevance of the selected features is necessary. The feature reduction process using PCA is not fully describe, how much variance is retained?

Response 4: We have expanded the Methods section to detail our feature selection process. Initially, 107 radiomic features were extracted. Features with an Intraclass Correlation Coefficient (ICC) greater than 0.8 were considered robust and selected for further analysis. Subsequently, PCA was applied to these robust features to reduce dimensionality.  PCA was applied to the extracted MRI radiomics features after Z-score normalization. The first two principal components were selected, accounting for 95% of the total variance. A scree plot and cumulative variance analysis were used to determine the optimal number of components, ensuring that the dimensionality reduction did not compromise critical biological information."

Round 2

Reviewer 3 Report

Comments and Suggestions for Authors

The authors have satisfactorily addressed my comments, and the changes added to the article highlight the results obtained, as well as the limitations and future directions that could be explored based on this work. I have no further comments.